# The aryl hydrocarbon receptor and FOS mediate cytotoxicity induced by *Acinetobacter baumannii*

Chun Kew[1,2], Cristian Prieto-Garcia [1], Anshu Bhattacharya[1,2], Manuela Tietgen [3,4], Craig R. MacNair[5], Lindsey A. Carfrae[5], João Mello-Vieira [1,2], Stephan Klatt [6], Yi-Lin Cheng [1,2,7], Rajeshwari Rathore[1], Elise Gradhand[8], Ingrid Fleming[6], Man-Wah Tan[5], Stephan Göttig [3], Volkhard A. J. Kempf[3] & Ivan Dikic [1,2,9,10] ✉

*Acinetobacter baumannii* is a pathogenic and multidrug-resistant Gram-negative bacterium that causes severe nosocomial infections. To better understand the mechanism of pathogenesis, we compare the proteomes of uninfected and infected human cells, revealing that transcription factor FOS is the host protein most strongly induced by *A. baumannii* infection. Pharmacological inhibition of FOS reduces the cytotoxicity of *A. baumannii* in cell-based models, and similar results are also observed in a mouse infection model. *A. baumannii* outer membrane vesicles (OMVs) are shown to activate the aryl hydrocarbon receptor (AHR) of host cells by inducing the host enzyme tryptophan-2,3-dioxygenase (TDO), producing the ligand kynurenine, which binds AHR. Following ligand binding, AHR is a direct transcriptional activator of the *FOS* gene. We propose that *A. baumannii* infection impacts the host tryptophan metabolism and promotes AHR- and FOS-mediated cytotoxicity of infected cells.

*Acinetobacter baumannii* is a Gram-negative bacterium that poses an increasing threat to public health based on its unusual ability to acquire multiple antibiotic resistance determinants. The bacterium is extremely versatile and can infect different parts of the human body, causing diseases such as pneumonia, meningitis, wound infections, and sepsis[1,2]. It is currently unclear how *A. baumannii* induces cytotoxicity in host cells. Unlike other pathogenic bacteria, *A. baumannii* does not rely on classic virulence effectors such as specialized cytotoxins and is often said to have a "persist and resist" pathogenic strategy[3]. *A. baumannii* shows an unusual degree of antibiotic resistance, and even pan-drug-resistant strains have been identified[4].

Infection with *A. baumannii* may, therefore, become untreatable with antibiotics in the future[2], and the World Health Organization (WHO) considers *A. baumannii* a "critical priority pathogen"[5], thus highlighting the urgent need for alternative therapies.

One such alternative approach is host-directed therapy[6]. Rather than targeting bacterial factors directly using antibiotics, host factors are modulated to overcome microbial toxicity or enhance pathogen clearance by augmenting the immune response[6,7]. Unlike antibiotics, host-directed interventions impose minimal selective pressure on pathogens, reducing the emergence of resistant strains[6]. Host-directed therapies against the tuberculosis pathogen *Mycobacterium*

[1]Institute of Biochemistry II, Faculty of Medicine, Goethe University, Frankfurt, Germany. [2]Buchmann Institute for Molecular Life Sciences, Goethe University, Frankfurt, Germany. [3]Institute for Medical Microbiology and Infection Control, Hospital of the Goethe University, Frankfurt, Germany. [4]University Center of Competence for Infection Control of the State of Hesse, Frankfurt, Germany. [5]Department of Infectious Diseases, Genentech Inc., 1 DNA Way, South San Francisco, CA, USA. [6]Institute for Vascular Signalling, Department of Molecular Medicine, CPI, Goethe University, Frankfurt, Germany. [7]Institute of Basic Medical Sciences, College of Medicine, National Cheng Kung University, Tainan, Taiwan. [8]Department of Pathology, Dr. Senckenberg Institute of Pathology, Goethe University, Frankfurt, Germany. [9]Fraunhofer Institute for Molecular Biology and Applied Ecology, Branch Translational Medicine and Pharmacology, Frankfurt, Germany. [10]Max Planck Institute of Biophysics, Frankfurt, Germany. ✉e-mail: dikic@biochem2.uni-frankfurt.de

*tuberculosis* have been evaluated successfully in preclinical studies and have entered clinical development[8]. Examples include the modulation of inflammation and the activation of intracellular antimicrobial defenses[9].

Host-directed therapy requires a clear understanding of the mechanisms underlying pathogen–host interactions. *A. baumannii* and other Gram-negative bacteria can interact with host cells via the secretion of nanoscale proteoliposomes known as outer membrane vesicles (OMVs) that bud from the bacterial outer membrane[10]. These structures carry diverse cargos and are necessary for pathogenesis in many bacteria. For example, *Porphyromonas gingivalis*[11–13] and *Escherichia coli*[14,15] utilize OMVs to transfer the virulence factors gingipain and hemolysin, respectively, into host cells. OMVs from *A. baumannii* cause mitochondrial damage, and this toxicity depends on the outer membrane protein OmpA[16].

It is unclear whether *A. baumannii* OMVs exert other effects on host cells, but one potential mechanism is the modulation of host transcription factors to favor pathogenesis. Indeed, many host transcription factors respond to both internal and external signals. For example, FOS (originally discovered as a proto-oncogene) responds to various growth factors and electrochemical signals as well as pathogens[17–20]. FOS dimerizes with members of the JUN family to form AP-1, which binds directly to DNA as a transcriptional activator. In the context of bacterial infection, the activation of FOS is induced by the MAP kinase signaling pathway and stimulates the production of cytokines[17–20]. Whether FOS activity contributes to other aspects of host-pathogen interactions, such as the cytotoxicity induced by bacteria, is unclear.

Pathogens can also interfere with host metabolism. For example, the aryl hydrocarbon receptor (AHR) is a transcription factor activated by highly diverse ligands such as the environmental pollutant 2,3,7,8-tetrachlorodibenzo-*p*-dioxin (TCDD), plant-derived indoles, and the products of tryptophan catabolism such as kynurenine. Activated AHR binds to aryl hydrocarbon receptor nuclear translocator (ARNT) and xenobiotic response elements (XREs) in target gene promoters to regulate transcription[21]. The target genes include those involved in the xenobiotic response, which helps to metabolize AHR ligands. Interestingly, AHR plays a key role in the regulation of immunity, and the activation of AHR is generally regarded as immunosuppressive[22,23]. Bacterial pathogens may, therefore, target AHR signaling to manipulate host physiology and immunity, thus favoring pathogenesis.

Here we demonstrate that FOS and AHR play an important role in *A. baumannii* pathogenesis. We found that a strong induction of FOS is a characteristic feature of *A. baumannii* infection. The pharmacological inhibition of FOS specifically prevented cell death induced by *A. baumannii* in cell-based models and led to a reduction of severity in a mouse model of infection. Intriguingly, we found that bacterial OMVs are potent activators of AHR, which directly induces *FOS* expression. Both the internalization of OMVs and bacterial lipid A synthesis are required for the activation of AHR. Furthermore, the host enzyme tryptophan-2,3-dioxygenase (TDO) is needed for the activation of AHR by OMVs. Exposure to OMVs swiftly activated TDO and the production of endogenous kynurenine in host cells. Together, these findings reveal molecular mechanisms of *A. baumannii* pathogenesis and suggest potential targets for host-directed therapies.

## Results

### *A. baumannii* infection strongly induces the expression of FOS

To better understand the host response to *A. baumannii* infection, we carried out a comparative proteomics analysis of A549 cells infected with *A. baumannii* (ATCC 19606) and mock-infected controls (Fig. 1a and Supplementary Data 1). A549 cells originate from lung epithelium and are often used in *A. baumannii* infection experiments[16,24,25]. The transcription factor FOS was the most strongly upregulated protein (Fig. 1b). FOSB, another member of the FOS family, was also

significantly upregulated (Supplementary Data 1). Gene ontology (GO) analysis revealed an enrichment of proteins involved in mitochondrial functions among the upregulated proteins (Supplementary Fig. 1a), consistent with previous findings showing the damaging effects of *A. baumannii* on mitochondria[16]. In contrast, response to inorganic substances, cell cycle and regulation of cell death were enriched among the downregulated proteins (Supplementary Fig. 1a). The upregulation of FOS was confirmed using different laboratory strains (ATCC 19606 and 17978) and clinical strains (CDC00035, CDC00036, CDC00037[26] and FDA-CDC AR-BANK#0280 (#0280)) of *A. baumannii* (Fig. 1c), and in human monocyte-derived macrophages (MDMs) (Fig. 1d). We also observed a moderate induction of JUN (Supplementary Fig. 1b and Supplementary Data 1). Intriguingly, FOS was induced more strongly by *A. baumannii* than the other Gram-negative pathogenic bacteria *E. coli* or *Pseudomonas aeruginosa*, whereas the activation of NF-κB was comparable for all three bacteria, as indicated by the degradation of IκB (Fig. 1e). The induction of FOS by *A. baumannii* is therefore more potent than that by *E. coli* and *P. aeruginosa*. Induction occurred at the transcriptional level, as shown by the higher *FOS* transcript levels measured by qRT-PCR (Fig. 1f and Supplementary Fig. 1c). Consistent with the results from cell culture, we observed the strong induction of FOS in the spleen, liver and lung tissues of mice infected with the *A. baumannii* clinical isolate FDA-CDC AR-BANK#0280 (#0280) (Fig. 1g). These results reveal that the unusual strong induction of FOS is a major host response to *A. baumannii* infection in vitro and in vivo.

### Inhibition of FOS alleviates the cytotoxicity of *A. baumannii* infection

Given that FOS is the most highly induced protein in *A. baumannii*-infected cells (Fig. 1), we tested whether FOS contributes to the cytotoxicity induced by *A. baumannii* infection using pharmacological inhibitors (Fig. 2a). T5224 is a FOS inhibitor with low in vivo toxicity and good bioavailability[27] and the only selective FOS inhibitor used in human clinical trials[28]. Interestingly, T5224 significantly attenuated the cytotoxicity induced by *A. baumannii* infection in different cells, including A549, lung fibroblast MRC-5, skin fibroblast BJ, keratinocyte HaCaT, and mouse macrophage-like RAW264.7 cells (Fig. 2b) but it did not affect the release of lactate dehydrogenase (LDH) from A549 cells in the absence of infection (Supplementary Fig. 2a, b). We observed the same attenuation of cytotoxicity by T5224 in A549 cells infected with different clinical strains of *A. baumannii* (Fig. 2c), suggesting this phenomenon is also relevant for clinical strains. SR11302 is a FOS inhibitor with no structural similarity to T5224[29]. However, like T5224, SR11302 also inhibited cell death induced by *A. baumannii* (Supplementary Fig. 2c), thus confirming the FOS-specific effect. T5224 did not rescue cell death induced by *P. aeruginosa*, indicating that the effect is also pathogen-specific (Supplementary Fig. 2d). The FOS inhibitors did not have any effect on the growth of *A. baumannii* (Supplementary Fig. 2e), indicating that the differences in cytotoxicity were not due to inhibition of the bacteria.

Because FOS is a transcription factor, we next investigated whether transcripts involved in cell death regulation are affected by T5224 treatment. We found that the expression of the anti-apoptotic genes *BIRC3*, *BCL3*, and *CDK5* was significantly enhanced by T5224 treatment during *A. baumannii* infection (Fig. 2d). Conversely, T5224 treatment suppressed the expression of the pro-apoptotic genes *BNIP3* and *BIM* in infected cells (Fig. 2e). Based on published data, these cell death regulators are known targets of FOS[30,31]. Furthermore, T5224 treatment reduced the cleavage of caspase 9 and PARP triggered by *A. baumannii* infection in A549 cells (Fig. 2f), suggesting that apoptosis is suppressed. We observed no significant change in cytokine levels during T5224 treatment (Supplementary Fig. 2f), suggesting that inflammation does not play a major role in FOS-mediated cytotoxicity. We therefore concluded that FOS inhibition specifically rescues apoptosis induced by *A. baumannii* infection.

T5224 appears to be a suitable candidate of host-targeting therapeutic for *A. baumannii* infection, so we tested the efficacy of T5224 in a mouse infection model (Fig. 2g). Wild-type BALB/c mice were intraperitoneally (i.p.) infected with the clinical isolate strain FDA-CDC AR-BANK#0280 (#0280), which is highly virulent and extensively drug-resistant[32]. Intraperitoneal infection in mice causes peritonitis and dissemination of bacteria and leads to systemic disease[33–35]. We chose to administrate T5224 both before and after infection since this infection model progresses rapidly (Supplementary Fig. 2i). A pre-infection treatment might be necessary for the maximal efficacy in this particular model. At 8 hours post-infection (hpi), we observed high levels of bacteria in the spleen, liver, and lung regardless of the treatment group, confirming bacterial dissemination (Supplementary Fig. 2g). Interestingly, mice treated with T5224 had lower levels of cleaved caspase 9 in their organs (Fig. 2h), which indicates a reduction of cell death, mirroring the phenomenon we observed in cells (Fig. 2f). Intriguingly, histological examination revealed a reduction of neutrophils and fibrin resembling an acute

peritonitis on the serosal surface of liver and spleen of infected animals treated with T5224 (Supplementary Fig. 2h). This indicates a less severe peritoneal inflammation. We did not observe any statistically significant increase of survival in infected mice treated with T5224 (Supplementary Fig. 2i). All in all, these results show that FOS inhibition reduces the cytotoxicity due to *A. baumannii* infection, both in vitro and in vivo.

## Bacterial OMVs trigger the induction of FOS

To investigate the molecular mechanism of FOS regulation in infected cells, we tested the effect of heat-inactivated bacteria. This abolished FOS induction but had little effect on NF-κB activation (Fig. 3a and Supplementary Fig. 3a), suggesting that bacterial activity, not just pathogen-associated molecular patterns (PAMPs), is required for FOS induction. Because OMV production is an active process that is required for bacterial pathogenesis[14,16,36,37], we set up transwell plates in which A549 cells were separated from *A. baumannii* cells by a 0.4-μm filter, preventing direct contact but allowing the exchange of secreted

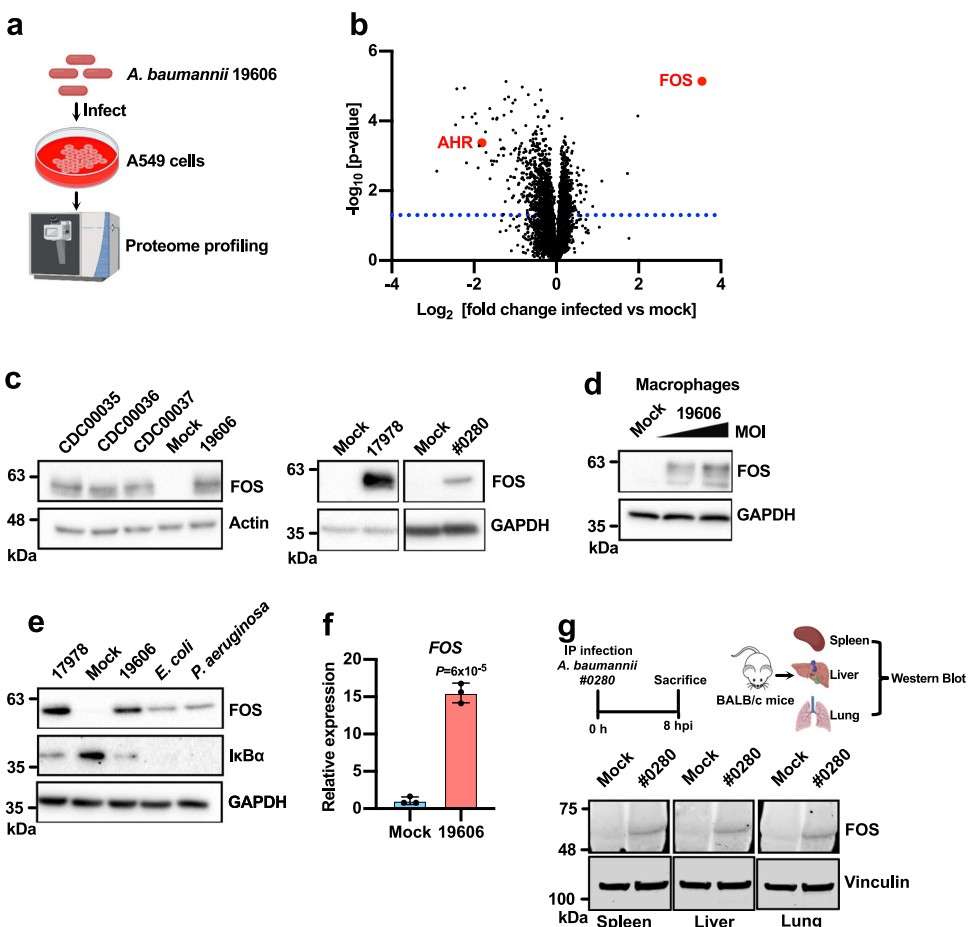

**Fig. 1 | *A. baumannii* infection strongly induces the expression of FOS.**
**a** Schematic representation of the proteomics experiment. Created with BioRender.com released under a Creative Commons Attribution-NonCommercial-NoDerivs 4.0 International license (https://creativecommons.org/licenses/by-nc-nd/4.0/deed.en). **b** Volcano plot of the proteomics data from A549 cells infected with *A. baumannii* ATCC 19606. Cells were harvested at 6 h post-infection (hpi). **c** Western blot analysis of A549 cells infected with *A. baumannii* showing the abundance of FOS, with actin or GAPDH for normalization (3 hpi). **d** Western blot analysis of MDM cells infected with *A. baumannii* (MOI 50 or 200) showing the abundance of FOS, with GAPDH for normalization (3 hpi). **e** Western blot analysis of A549 cells exposed to different bacteria: *A. baumannii*, enteropathogenic *E. coli* O127:H6 and *P. aeruginosa* ATCC 27853 (3 hpi). **f** Quantification of *FOS* mRNA levels

by qRT-PCR in A549 cells infected with *A. baumannii* (3 hpi). $n = 3$ independent experiments. Data are presented as mean values ±SD, *P*-values by unpaired two-tailed *t*-test. **g** The mouse infection challenge model. Wild-type BALB/c mice were injected intraperitoneally (i.p.) with $3 \times 10^3$ colony forming units (CFU) of *A. baumannii* clinical isolate FDA-CDC AR-BANK#0280 (#0280). The animals were euthanized 8 hpi and organs were harvested for western blot analysis. Three pairs of mock and infected animals were analyzed, and similar results were obtained. The experimental design illustration was created with BioRender.com released under a Creative Commons Attribution-NonCommercial-NoDerivs 4.0 International license (https://creativecommons.org/licenses/by-nc-nd/4.0/deed.en). **c**–**e** The experiments were repeated three times with similar results obtained. See also Supplementary Fig. 1. Source data are provided in the Source Data file.

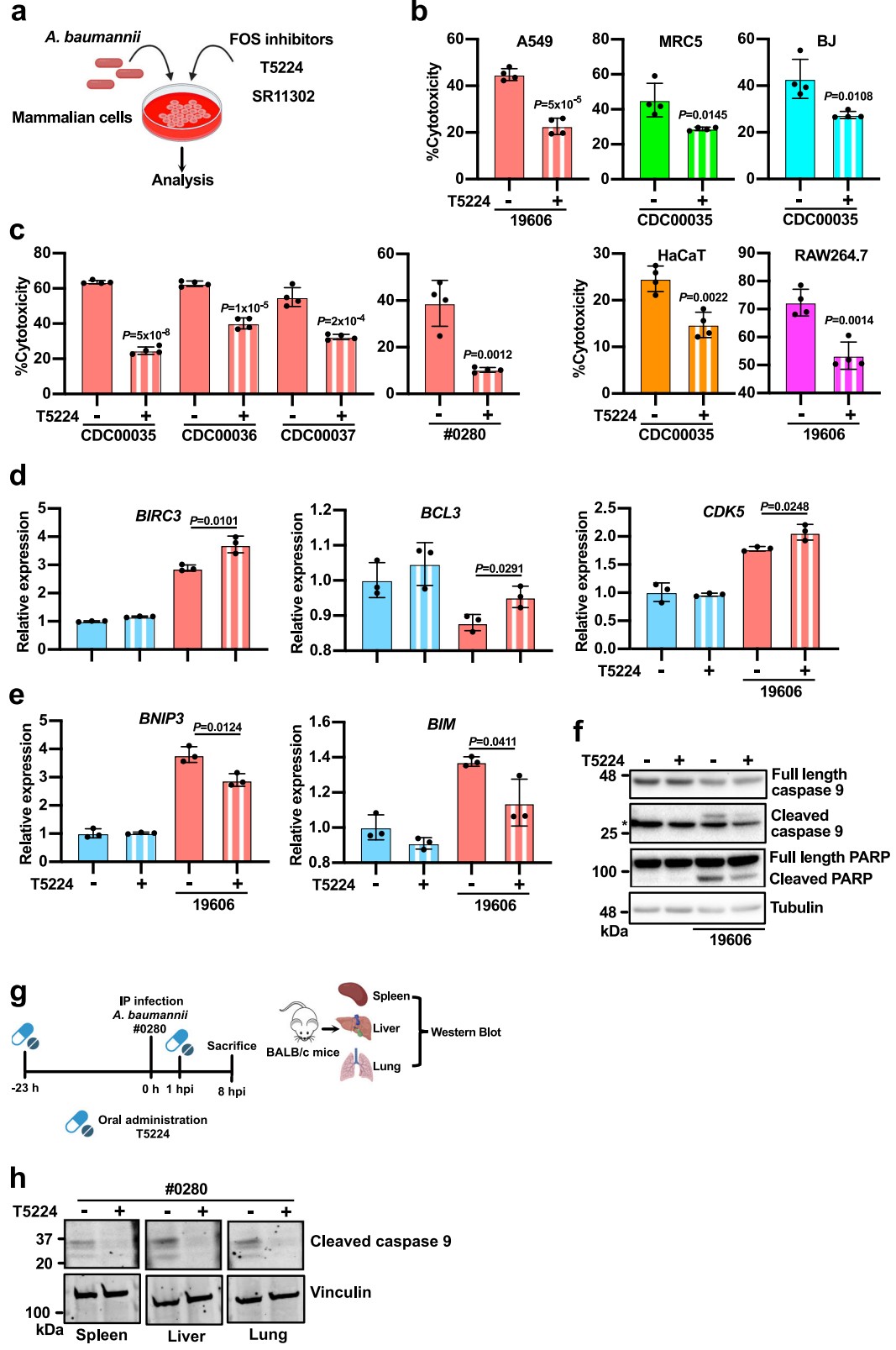

factors, such as OMVs. Under such conditions, *A. baumannii* was still able to induce FOS in the host cells (Fig. 3b and Supplementary Fig. 3b). Consistently, treating the cells with purified OMVs also resulted in the upregulation of FOS (Fig. 3c). The involvement of OMVs was supported by the use of OMV production modulators. Sub-inhibitory concentrations of colistin destabilize the bacterial outer membrane and promote the production of OMVs[38]. Accordingly, the presence of colistin during

*A. baumannii* infection further augmented FOS induction (Fig. 3d), whereas the OMV inhibitor BB-Cl-amidine[39,40] had the opposite effect (Fig. 3e).

The outer membrane protein OmpA is a major virulence factor carried by OMVs[16]. To determine whether OmpA is required for the induction of FOS, we infected A549 cells with the *A. baumannii* Δ*ompA* mutant. Unexpectedly, we observed an even stronger

**Fig. 2 | Inhibition of FOS alleviates the cytotoxicity of *A. baumannii* infection.**
**a** Schematic representation of the FOS inhibition experiments. Created with
BioRender.com released under a Creative Commons Attribution-NonCommercial-
NoDerivs 4.0 International license (https://creativecommons.org/licenses/by-nc-
nd/4.0/deed.en). **b, c** Lactate dehydrogenase (LDH) release assay in A549 (24 hpi),
MRC-5, BJ, HaCaT (12 hpi), and RAW264.7 (24 hpi) cells infected with *A. baumannii*
and treated with the FOS inhibitors T5224 (100 μM). **d, e** Quantification of *BIRC3*,
*BCL3*, *CDK5*, *BNIP3*, and *BIM* mRNA levels by qRT-PCR in A549 cells infected with *A.
baumannii* (3 hpi) with or without T5224 treatment (100 μM). *n* = 3 independent
experiments. **f** Western blot analysis of A549 cells infected with *A. baumannii* (6
hpi) with or without T5224 treatment (100 μM); * indicates nonspecific bands. The
experiment was repeated three times with similar results obtained. **g** The mouse
infection challenge model with T5224 treatment. Wild-type BALB/c mice were

administrated orally with T5224 (250 mg/kg) or vehicle before and after injection
i.p. with 3 × 10³ CFU of *A. baumannii* clinical isolate FDA-CDC AR-BANK#0280
(#0280). The animals were euthanized 8 hpi. Created with BioRender.com released
under a Creative Commons Attribution-NonCommercial-NoDerivs 4.0 Interna-
tional license (https://creativecommons.org/licenses/by-nc-nd/4.0/deed.en).
**h** Western blot analysis of tissues harvested from the mouse infection model
described in (**h**). Three pairs of infected animals were analyzed, and similar results
were obtained. **b, c** *n* = 4. Cells were seeded in four different wells per group,
Treatment and measurement were performed independently for each well.
Experiments were repeated independently three times and similar results were
obtained. **b–e** Data are presented as mean values ±SD, *P*-values by unpaired two-
tailed *t*-test. See also Supplementary Fig. 2. Source data are provided in the Source
Data file.

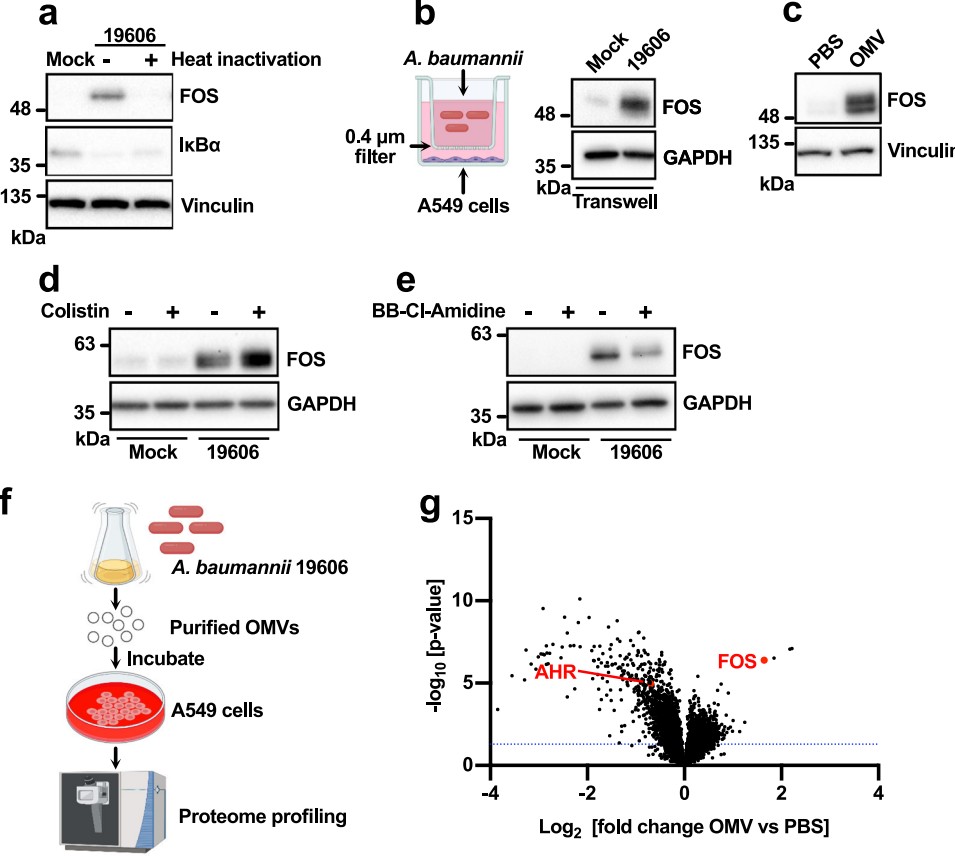

**Fig. 3 | Bacterial OMVs trigger the induction of FOS. a** Western blot analysis of
A549 cells exposed to live or heat-inactivated *A. baumannii* for 3 h. **b** Western blot
analysis of A549 cells infected with *A. baumannii* in a transwell setting (3 hpi). The
illustration of the transwell was created with BioRender.com released under a
Creative Commons Attribution-NonCommercial-NoDerivs 4.0 International license
(https://creativecommons.org/licenses/by-nc-nd/4.0/deed.en). **c** Western blot
analysis of A549 cells treated with purified *A. baumannii* OMVs (100 μg/mL for 3 h)
or PBS. **d, e** Western blot analysis of A549 cells infected with *A. baumannii* (3 hpi for
the colistin treatment group and 6 hpi for the BB-Cl-Amidine treatment group).

Colistin (0.2 μg/mL) and BB-Cl-Amidine (5 μM) modulate the production of OMVs.
**f** Schematic representation of the proteomics experiment. Created with BioR-
ender.com released under a Creative Commons Attribution-NonCommercial-
NoDerivs 4.0 International license (https://creativecommons.org/licenses/by-nc-
nd/4.0/deed.en). **g** Volcano plot of the proteomics data from A549 cells exposed to
*A. baumannii* OMVs (100 μg/mL for 3 h). **a–e** The experiments were repeated three
times with similar results obtained. See also Supplementary Fig. 3. Source data are
provided in the Source Data file.

induction of FOS in cells infected with this mutant (Supplementary
Fig. 3c, d), suggesting that FOS induction does not require OmpA.
This was confirmed in the transwell plates setting (Supplementary
Fig. 3e), indicating that secreted factors, such as OMVs, are involved.
Consistently, FOS inhibition by T5224 also limited cytotoxicity in
cells infected with the *ΔompA* mutant (Supplementary Fig. 3f), con-
firming that FOS-mediated cytotoxicity does not require OmpA. The
*ΔompA* mutant is known to produce more OMVs than normal
laboratory and clinical strains[41], which might explain the stronger
induction of FOS during infection.

To gain deeper insight into the effect of OMVs on host cells, we
repeated the earlier comparative proteomics analysis, this time using
A549 cells treated with OMVs purified from *A. baumannii* (Fig. 3f). As
observed in the infected cells (Fig. 1b), FOS was one of the proteins
most strongly induced by OMVs (Fig. 3g and Supplementary Data 2).
GO analysis showed that the upregulated proteins were enriched for
the establishment of protein localization, ribosome biogenesis, cel-
lular macromolecule localization, and RNA processing, whereas the
downregulated proteins were enriched for the regulation of the cell
cycle and cell division (Supplementary Fig. 3g).

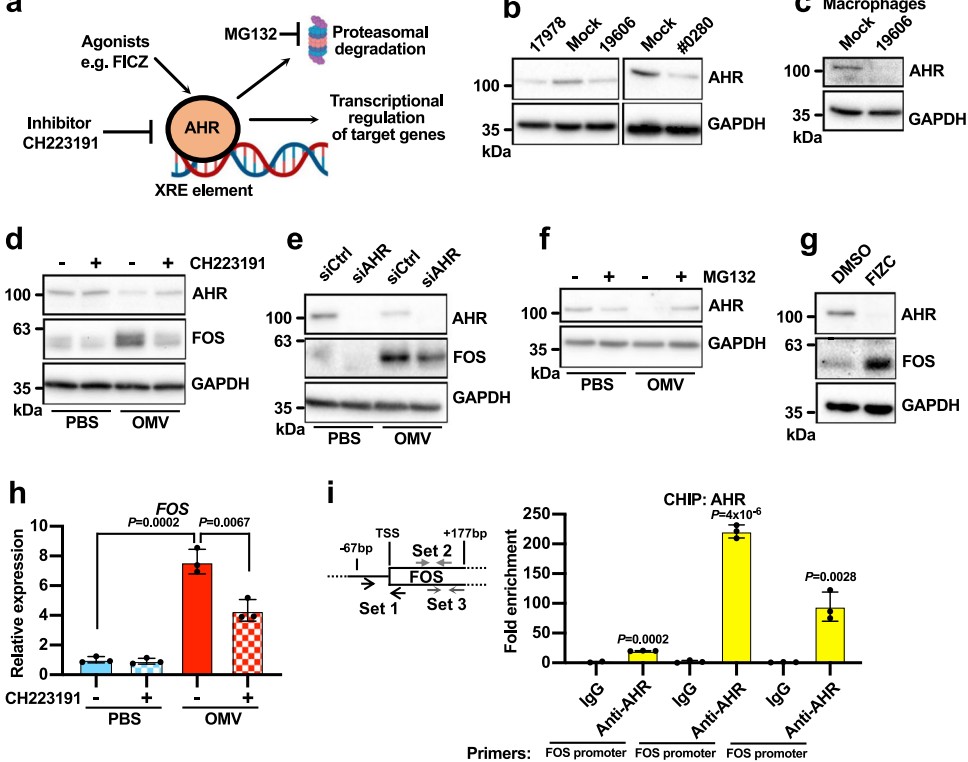

**Fig. 4 | FOS induction requires AHR. a** Schematic representation of the regulation of AHR in human cells. Created with BioRender.com released under a Creative Commons Attribution-NonCommercial-NoDerivs 4.0 International license (https://creativecommons.org/licenses/by-nc-nd/4.0/deed.en). **b, c** Western blot analysis of A549 (B) and MDM cells (C) infected with *A. baumannii* (3 hpi). **d** Western blot analysis of A549 cells treated with purified *A. baumannii* OMVs (100 μg/mL for 3 h) or PBS. CH223191 (10 μM) was added to the designated groups to inhibit AHR. **e** Western blot analysis of A549 cells treated with purified *A. baumannii* OMVs (100 μg/mL for 3 h) or PBS. The cells were transfected with the indicated siRNA 48 h before OMV treatment. **f** Western blot analysis of A549 cells treated with purified *A. baumannii* OMVs (100 μg/mL for 3 h) or PBS. MG132 (20 μM) was added to the designated groups to inhibit the proteasome. **g** Western blot analysis of A549 cells treated with FICZ (1 μM for 3 h). **h** Quantification of *FOS* mRNA levels by qRT-PCR in A549 cells exposed to *A. baumannii* OMVs (100 μg/mL for 3 h) and treated with CH223191 (10 μM). **i** ChIP-qPCR analysis of AHR. DNA associated with AHR in A549 cells was immunoprecipitated using an anti-AHR antibody. Enrichment of the *FOS* promoter region was confirmed by qPCR. The schematic on the left shows the position of the primers relative to the transcriptional start site (TSS) of *FOS*. (b-g) The experiments were repeated three times with similar results obtained. **h, i** *n* = 3 independent experiments. Data are presented as mean values +/- SD, P-values by unpaired two-tailed t-test. See also Supplementary Fig. 4. Source data are provided in the Source Data file.

## FOS induction requires AHR

Next, we investigated the downstream steps in the induction of FOS by OMVs. The receptor AHR was downregulated in both infected and OMV-treated cells (Figs. 1b and 3g). AHR is required for the expression of *FOS* in bone cells during osteoclastogenesis[42], and AHR is degraded by the proteasomal system following activation (Fig. 4a)[43]. Therefore, we hypothesized that AHR is required for the induction of FOS by OMVs. First, we confirmed the downregulation of AHR in infected A549 cells (Fig. 4b), infected MDM cells (Fig. 4c), and OMV-treated A549 cells (Fig. 4d). Blocking AHR activity using the chemical inhibitor CH223191 or siRNA significantly reduced the induction of FOS by OMVs (Fig. 4d, e). CH223191 treatment also prevented the degradation of AHR (Fig. 4d), indicating that it inhibits AHR activation. AHR is degraded by the proteasome following OMV exposure, as confirmed by the ability of the proteasomal inhibitor MG132 to prevent the AHR downregulation (Fig. 4f). Although OMV treatment led to rapid degradation of AHR in 3 h, the protein levels of FOS and CYP1A1, a major target of AHR, remained highly upregulated even after 6 h of OMV treatment (Supplementary Fig. 4), suggesting that the effects of AHR activation persist even after its degradation. Furthermore, the activation of AHR using 6-formylindolo(3,2-*b*)carbazole (FICZ) induced FOS protein expression (Fig. 4g). We concluded that AHR is the mediator of FOS expression induced by OMVs.

AHR is a transcription factor, so we tested whether the *FOS* gene is directly regulated by AHR. Inhibition of AHR repressed *FOS* transcript induction by OMVs (Fig. 4h). AHR also physically bound to the *FOS* promoter region in chromatin immunoprecipitation (ChiP)-qPCR experiments (Fig. 4i). Based on these results, we concluded that AHR is a direct transcriptional regulator of FOS.

## OMVs from *A. baumannii* activate AHR

Given that AHR is needed for the induction of FOS by OMVs (Fig. 4), it is likely that OMVs activate AHR. To test this hypothesis, we used a luciferase reporter system driven by the xenobiotic response element (XRE), which is the target site for activated AHR[43]. As expected, purified OMVs from *A. baumannii* potently activated the reporter in an AHR-dependent manner (Fig. 5a). OMVs facilitate the intracellular delivery of bacterial factors, and endocytosis is believed to be the major route for OMVs to enter mammalian cells[10]. To determine whether the activation of AHR by OMVs requires internalization, we repeated the XRE luciferase assay in the presence of cytochalasin B, dynasore, and bafilomycin A1. Cytochalasin B inhibits the assembly of actin filament networks[44] whereas dynasore inhibits dynamin-mediated membrane scission[45]. Both processes are necessary for the endocytosis of OMVs (Fig. 5b)[10]. The treatment of A549 cells with cytochalasin B or dynasore significantly inhibited luciferase reporter gene activation induced by OMVs

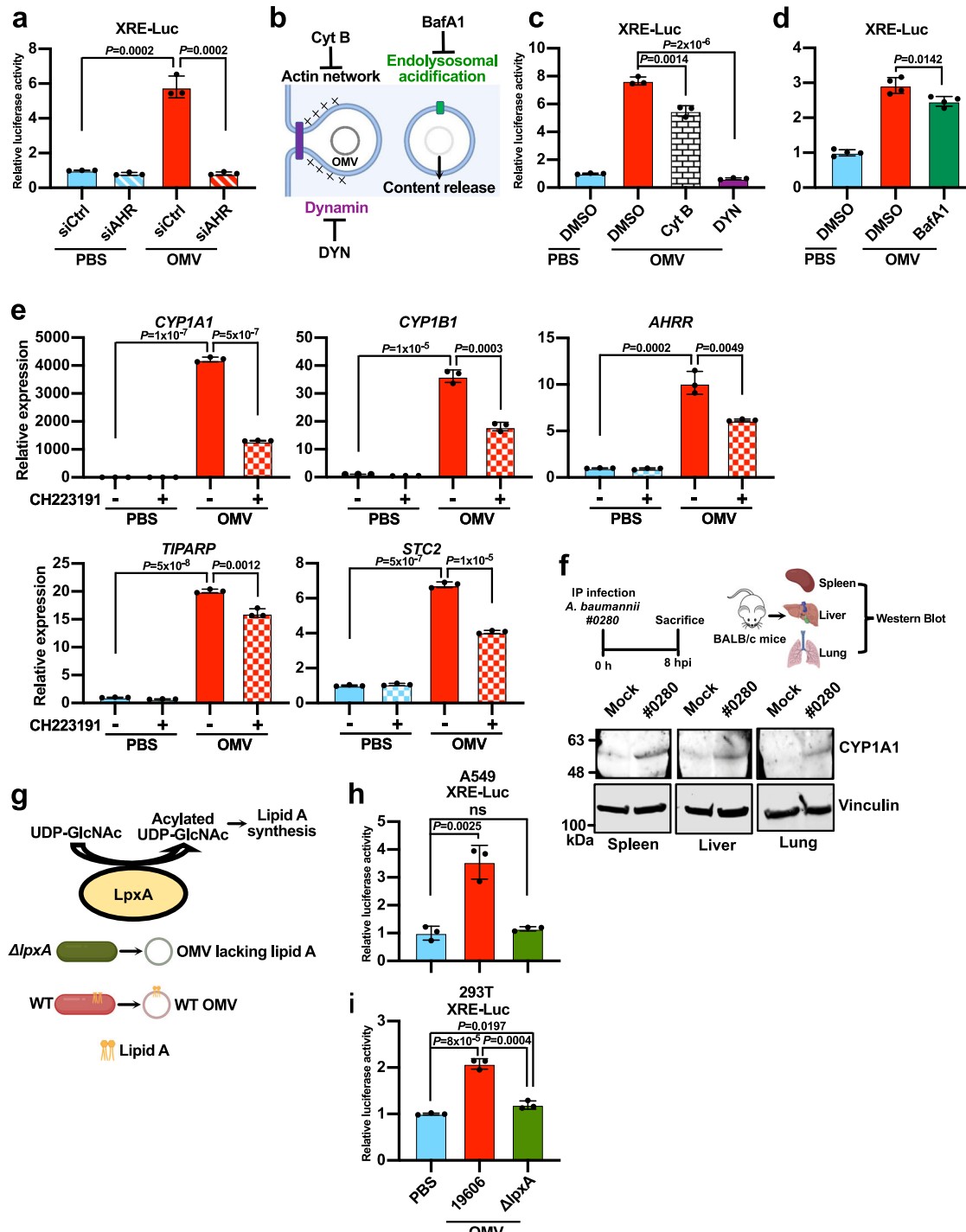

(Fig. 5c), indicating that OMVs must be taken up to activate AHR. The release of cargo from internalized OMVs can be triggered by endolysosomal acidification (Fig. 5b), and this process is inhibited by bafilomycin A1[14,46]. The addition of bafilomycin A1 also slightly inhibited luciferase reporter gene activation induced by OMVs (Fig. 5d). Endolysosomal acidification might play a secondary role in the activation of AHR as the effect of bafilomycin A1 was relatively small. In addition, AHR target genes such as *CYP1A1*, *CYP1B1*, *AHRR*, *TIPARP*, and *STC2* were strongly induced by OMV treatment, as determined by qRT-PCR, and the induction was partially abolished by the AHR inhibitor CH223191 (Fig. 5e). The induction of AHR target genes was also observed in cells infected with *A. baumannii* (Supplementary Fig. 5a, b). Accordingly, organs from mice infected

with *A. baumannii* contained elevated levels of CYP1A1 protein (Fig. 5f), indicating that AHR is activated in vivo following infection with *A. baumannii*. Based on these observations, we concluded that internalized OMVs activate AHR.

We also tested whether the FOS inhibitors T5224 and SR11302 are inducers of AHR since AHR is known to be promiscuous[21]. The XRE luciferase was potently activated by FICZ, a known activator of AHR, while there was no significant induction by T5224 or SR11302 (Supplementary Fig. 5c). FICZ also potently induced the transcription of *CYP1A1*. On the other hand, T5224 resulted in a very small induction of CYP1A1 while SR13302 did not have an effect (Supplementary Fig. 5d). All in all, these data indicate that T5224 and SR11302 are not relevant activators of AHR.

**Fig. 5 | OMVs from *A. baumannii* activate AHR. a** AHR activity reporter (XRE-Luc) assay in A549 cells exposed to OMVs (100 µg/mL for 3 h). Cells were transfected with the indicated siRNA 48 h before OMV treatment. **b** Schematic showing the key steps of OMV endocytosis and the corresponding inhibitors. Created with BioRender.com released under a Creative Commons Attribution-NonCommercial-NoDerivs 4.0 International license (https://creativecommons.org/licenses/by-nc-nd/4.0/deed.en). **c, d** AHR activity reporter (XRE-Luc) assay in A549 cells exposed to OMVs (20 µg/mL for 3 h) in the presence of the following chemicals: DMSO (solvent control), CytB (cytochalasin B, inhibitor of actin polymerization), DYN (dynasore, inhibitor of dynamin) and BafA1 (bafilomycin A1, inhibitor of endolysosomal acidification). **e** Quantification of AHR target gene expression (*CYP1A1*, *CYP1B1*, *AHRR*, *TIPARP* and *STC2*) by qRT-PCR in A549 cells exposed to OMVs (100 µg/mL for 3 h). CH223191 (10 µM) was added to inhibit AHR. *n* = 3 independent experiments. **f** The mouse infection challenge model. Wild-type BALB/c mice were injected i.p. with 3 ×10³ CFUs of *A. baumannii* clinical isolate #0280 followed by western blot analysis of organs for the detection of CYP1A1, with vinculin for normalization. 3 pairs of

mock and infected animals were analyzed, and similar results were obtained. The illustration was created with BioRender.com released under a Creative Commons Attribution-NonCommercial-NoDerivs 4.0 International license (https://creativecommons.org/licenses/by-nc-nd/4.0/deed.en). **g** Schematic showing the function of LpxA in lipid A synthesis. OMVs produced by the *ΔlpxA* mutant strain of *A. baumannii* lack lipid A. Created with BioRender.com released under a Creative Commons Attribution-NonCommercial-NoDerivs 4.0 International license (https://creativecommons.org/licenses/by-nc-nd/4.0/deed.en). **h, i** AHR activity reporter (XRE-Luc) assay in A549 (H) and HEK 293 T (I) cells exposed to OMVs (100 µg/mL for 3 h) isolated from ATCC 19606 or the *A. baumannii ΔlpxA* mutant (lipid A synthesis mutant). **a, c, h, i** *n* = 3. **d** *n* = 4. Cells were seeded in 3 or 4 different wells per group, Treatment and measurement were performed independently for each well. Experiments were repeated independently three times and similar results were obtained. **a, c, d, e, h, i** Data are presented as mean values +/- SD, P-values by unpaired two-tailed t-test. See also Supplementary Fig. 5. Source data are provided in the Source Data file.

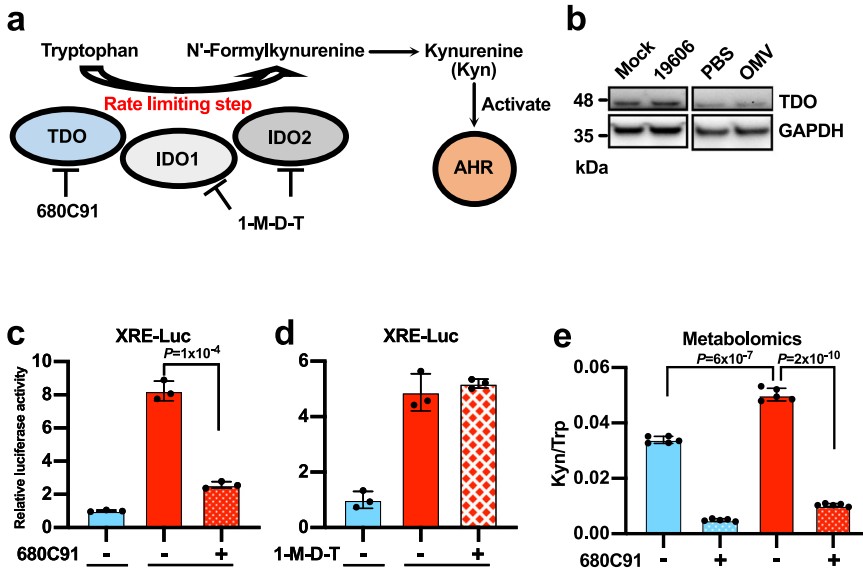

**Fig. 6 | OMVs activate AHR by modulating host tryptophan metabolism.**
**a** Schematic showing the simplified tryptophan catabolic pathway in human cells. **b** Western blot analysis of A549 cells exposed to *A. baumannii* or its OMVs (100 µg/mL for 3 h). The experiment was repeated three times with similar results obtained. **c, d** AHR activity reporter (XRE-Luc) assay in A549 cells exposed to OMVs (100 µg/mL for 3 h) in the presence of the TDO inhibitor 680C91 (1 µM) or the IDO inhibitor 1-M-D-T (100 µM). **e** Ratio of kynurenine to tryptophan in A549 cells exposed to

OMV (100 µg/mL) and 680C91 (1 µM) for 1 h. Metabolite levels were measured by LC-MS. *n* = 5 independent experiments. **c, d** *n* = 3. Cells were seeded in three different wells per group, treatment and measurement were performed independently for each well. Experiments were repeated independently three times and similar results were obtained. **c–e** Data are presented as mean values ±SD, P-values by unpaired two-tailed *t*-test. See also Supplementary Fig. 6. Source data are provided in the Source Data file.

Lipopolysaccharide (LPS) is an abundant component of OMVs. LPS isolated from *E. coli* can activate AHR[22]. *A. baumannii* does not synthesize LPS but produces a similar molecule known as lipooligosaccharide (LOS)[3]. Lipid A is an obligatory component of both LPS and LOS and is the main PAMP sensed by host cells[47]. Although lipid A synthesis is essential for the survival of most bacteria, *A. baumannii* mutants lacking lipid A remain viable[48,49]. We exploited this unique property to investigate the role of lipid A in the activation of AHR. Interestingly, while OMVs from wild-type bacteria efficiently activated AHR, equal amounts of OMVs isolated from the lipid A-deficient mutant *ΔlpxA* (Fig. 5g) were unable to activate AHR (Fig. 5h), demonstrating that lipid A synthesis is required for OMVs to activate AHR. To determine whether host receptors for lipid A are also required, we repeated the AHR reporter gene experiments using HEK 293T cells, which lack functional receptors for lipid A (including TLR4 and caspase 4/5) and are therefore unresponsive to lipid A stimulation[50,51]. Surprisingly, OMVs still activated AHR in these cells in a lipid A-dependent

manner (Fig. 5i). Neither purified lipid A nor cell lysate from *A. baumannii* was sufficient to activate AHR, highlighting the importance of OMVs as a bacterial factor (Supplementary Fig. 5e). Collectively, these results suggest a novel AHR activation mechanism which is dependent on OMVs and bacterial lipid A synthesis.

**OMVs activate AHR by modulating host tryptophan metabolism**
One way that OMVs could activate AHR is to stimulate the endogenous production of AHR ligands. The major AHR ligands produced in human cells are products of tryptophan catabolism, and this pathway is controlled by three different enzymes (IDO1, IDO2 and TDO), each responsible for catalyzing the same rate-limiting step (Fig. 6a)[52]. IDO1 and IDO2 are unlikely to be involved because they are not expressed at detectable levels in A549 and Huh-7 cells (Supplementary Fig. 6a–d). In contrast, we readily detected TDO (Fig. 6b and Supplementary Fig. 6e), suggesting this is the main enzyme that breaks down tryptophan in A549 and Huh-7 cells. Consistently, the specific TDO chemical inhibitor

680C91 blocked the activation of AHR in response to OMVs (Fig. 6c) whereas 1-methyl-D-tryptophan (1-M-D-T), a specific inhibitor of IDOs, had no significant effect (Fig. 6d). TDO levels were not altered by *A. baumannii* infection or OMV treatment (Fig. 6b) although the transcript levels were reduced slightly (Supplementary Fig. 6f). TDO is strongly expressed in the liver[52], so we tested whether OMVs have the same effect on hepatocyte-derived Huh-7 cells[53]. These responded to OMVs in the same way as A549 cells (Supplementary Fig. 5d, e, g), suggesting the whole pathway is conserved in liver cells.

Finally, we measured the levels of tryptophan metabolites in cells treated with OMVs by liquid chromatography-mass spectrometry (LC-MS). We found that kynurenine, the downstream product of TDO and the main endogenous AHR ligand, was elevated significantly following OMV exposure for just 1 h. Furthermore, the addition of 680C91 largely suppressed the induction of kynurenine caused by OMV treatment (Fig. 6e). Taken together, these results suggest that *A. baumannii* OMVs do not induce the expression of the host tryptophan catabolic enzymes, but specifically activate TDO post-translationally, which in turn generates ligands that activate AHR.

## Discussion

We discovered a mechanism of cytotoxicity induced by *A. baumannii* infection which involves the interplay between invading bacteria, the host transcription factor FOS, and host tryptophan metabolism (Supplementary Fig. 7). The potential contribution of host transcription factors to the cytotoxicity induced by bacteria has been largely neglected. *A. baumannii* infections result in a stronger induction of FOS when compared to the two other similar bacterial pathogens that we tested, and FOS activity is required for full virulence of *A. baumannii* in vitro and in vivo. The induction of FOS is mediated at least in part by the take-up of secreted bacterial OMVs, which increase the activity of the host enzyme TDO to promote tryptophan catabolism and the accumulation of kynurenine. This metabolite then activates AHR, a direct transcriptional activator of *FOS*. Importantly, this pathway is easy to block with available drug candidates, making it an attractive target for host-directed therapeutics.

FOS is a regulator of cell death[54]. The involvement of FOS in innate immunity and neoplasia has also been documented[17,20,55]. However, whether FOS plays a role in the cell death caused by bacterial infections remains unknown. We found that *A. baumannii* infections cause a high upregulation of FOS, whose activity results in host cell death (Fig. 2). FOS inhibition prevented cell death induced by *A. baumannii* but not *P. aeruginosa* (Fig. 2 and Supplementary Fig. 2d). It should be noted that in the current study, we compared only a small number of bacterial strains. In the future, it would be interesting to investigate more bacteria with regard to the induction of FOS and the potential prevention of cytotoxic effects by FOS inhibitors. The FOS inhibitor T5224 has already been used in human clinical trials for other diseases, such as arthritis[28]. The repurposing of T5224 for bacterial infections would be more convenient than de novo drug development.

We found that OMVs are potent activators of AHR, which can directly induce the expression of *FOS* (Figs. 4 and 5). The role of OMVs in bacterial pathogenesis has recently gained more prominence[56]. Our findings represent an important addition to the arsenal of ways in which OMVs usurp host physiology. AHR activation directs T-cell maturation and differentiation, which leads to the suppression of immunity and inflammation[57]. AHR is involved in diverse pathogenic conditions, including graft vs host disease, autoimmune diseases, neurodegenerative diseases, and cancer[21], which are potentially influenced by bacterial OMVs. This is important because OMVs are stable in vivo and can penetrate biological barriers that are non-permissive to intact bacterial cells, such as the blood–brain barrier[58]. This would enable the bacteria to influence the physiology of distant organs.

Our results indicate that OMVs activate AHR indirectly via host tryptophan metabolism, specifically the catabolic enzyme TDO but not

IDOs (Fig. 6). Tryptophan degradation is enhanced during inflammation to generate immunoregulatory ligands for AHR, such as kynurenine[22], in a process mediated by IDOs[52]. IDOs are normally found at basal levels but their expression can be induced in a wide range of tissues by IFNγ during inflammation[52]. In contrast, TDO is believed to be responsible for the maintenance of homeostatic levels of tryptophan and kynurenine[52]. TDO is expressed at stable levels mainly in the liver[52], but also in the lung[59], kidney[60,61], and brain[62,63]. Consistently, we did not detect IDO expression in A549 or Huh7 cells where TDO was readily detectable (Fig. 6b and Supplementary Fig. 6a–f). In agreement with these observations, AHR activation by OMVs is also dependent on TDO but not IDOs (Fig. 6c, d). Contrary to the current beliefs of the predominate role of IDOs during inflammation, the expression or activity of IDOs are not altered by OMVs. Rather, OMVs activate TDO without affecting its expression level (Fig. 6b and Supplementary Fig. 6e), indicating a novel mechanism for enhanced kynurenine production via a post-translational activation of TDO by OMVs. The proposed regulation of the liver-enriched enzyme TDO by bacterial OMVs is fascinating because OMVs accumulate in the liver in vivo[12,64,65]. We hypothesized that high levels of OMVs in the liver might result in the activation of TDO and AHR in hepatocytes, thus influencing hepatic functions. An increase of TDO activity can lead to the depletion of tryptophan, which triggers the activation of GCN2[66] and thus the suppression of immune responses[67–69], which could assist *A. baumannii* infections. The precise molecular mechanism used by OMVs to induce TDO activity should be investigated in future studies.

Lipid A is the main PAMP of Gram-negative bacteria and their OMVs[70]. We found that the activation of AHR by OMVs requires bacterial lipid A synthesis and internalization (Fig. 5), suggesting that intracellular lipid A may be responsible for the OMV-mediated activation of the TDO–AHR signaling pathway. Alternatively, the full activation of AHR might require other components of OMVs. Cellular recognition of lipid A is mediated by TLR4 and caspase 4/5 (caspase 11 in mice)[70]. Lipid A and OMVs have been shown to induce potent host responses in cells expressing TLR4 and/or caspase 4/5/11 whereas cells devoid of functional TLR4 and caspase 4/5/11, such as HEK293T cells, are unresponsive to lipid A[50,51]. Surprisingly, OMVs still induce robust lipid A-dependent AHR activation in HEK 293T cells (Fig. 5i). This suggests the lipid A-dependent activation of AHR is independent of TLR4 and caspase 4/5/11. Alternatively, a lack of LpxA may alter the structures or components of OMVs in addition to lipid A, which might also contribute to the loss of AHR activation. Given the swiftly increasing kynurenine production in host cells (Fig. 6e, samples harvested 1 hour after OMV treatment), we speculate that TDO or its interacting partners might directly interact with OMVs.

This study provides the first promising results regarding the reduction of the severity of intraperitoneal *A. baumannii* infection in a mouse model using FOS inhibitors. However, a more thorough investigation is needed to further consolidate the potential of FOS inhibitors as host-directed-therapeutics against bacterial infections. For example, it would be useful to include pneumonia models in future studies, which might be more representative for human infections.

In conclusion, our study facilitates the future exploration of the intriguing connection between bacterial OMVs, host metabolism, and cell death pathways. Given the current antibiotic resistance crisis, we propose further in vivo investigation of FOS inhibitors in bacterial infections. This represents a direction for developing therapies alternative to traditional antibiotics.

## Methods
### Ethical regulation
The animal studies were reviewed and approved by the ethic committee PDST IACUC (Pharmacology Discovery Services Taiwan Institutional Animal Care and Use Committee). We confirm that the animal experiments complied with all relevant ethical regulations.

## Mammalian cell culture

A549, RAW264.7, MRC-5, BJ, and HEK 293 T cells were obtained from the American Type Culture Collection (ATCC). Human monocyte-derived macrophages (MDMs) were differentiated from peripheral blood mononuclear cells (PBMCs) collected from a local blood donation center. PBMCs were washed with PBS and incubated in RPMI 1640 medium supplemented with 2.5% heat-inactivated human serum for ~7 days, allowing them to differentiate. All other cells were cultivated in RPMI 1640 (A549 cells) or Dulbecco's modified Eagle's medium (DMEM) (RAW264.7, MRC-5, BJ, HaCaT, Huh7, and HEK 293T cells) supplemented with 10% fetal calf serum. All cells were cultured at 37 °C in a humid 5% carbon dioxide atmosphere. All cells used in this study were tested negative for mycoplasma contamination.

## Bacterial culture

The *A. buamannii ΔlpxA* mutant[26] and the corresponding ATCC 19606 wild-type strain were cultured in tryptic soy broth (TSB) and on tryptic soy agar (TSA). The following strains were cultured in lysogeny broth (LB) and on LB agar: *E. coli* O127:H6, *P. aeruginosa* ATCC 21853, *A. baumannii* ATCC 17978, 19606 (when not serving as the wild-type strain for the *ΔlpxA* mutant), FDA-CDC AR-BANK#0280, CDC00035, CDC00036 and CDC00037. FDA-CDC AR-BANK#0280 was cultivated in brain heart infusion (BHI) broth for the mouse experiments. Bacteria were cultivated at 37 °C following standard procedures.

## Mouse housing and husbandry

Experiments on live animals were carried out by Pharmacology Discovery Services Taiwan, Ltd. (partner of Eurofins Discovery) as an experimental paid service (SOW # ODS112-01006934). All aspects of animal experiments, including housing, treatments, and disposal, adhered to the "Guide for the Care and Use of Laboratory Animals: Eight Edition"[71] and were carried out in an Association for Assessment and Accreditation of Laboratory Animal Care (AAALAC)-accredited animal facility.

All animal experiments involved BALB/c immunocompetent female mice at 5-6 weeks of age, which were closely monitored for health on a daily basis. The animals were maintained in a controlled temperature (20 – 24 °C) and humidity (30% – 70%) environment with 12-h light/dark cycles. Following infection with *A. baumannii*, animals were monitored for mortality and health at least twice per day. When they reached early humane endpoints, showed signs of pain, or became moribund, they were immediately euthanized. The responsible veterinarian had full authority to remove animals from the study and/or perform euthanasia prior to the experimental endpoint in order to prevent unacceptable levels of pain or distress.

## In vitro model of bacterial infection

Mammalian cells were seeded on the day before infection to generate confluent monolayers for infection experiments. Saturated bacterial cultures were diluted 1:10 in fresh LB or TSB (for experiments involving the *ΔlpxA* mutant). The diluted cultures were then grown at 37 °C for ~2 h shaking at 180 rpm to reach an $OD_{600}$ of ~1.0. Bacterial cells were then pelleted by centrifugation and washed twice with RPMI or DMEM. After resuspending the bacteria in cell culture medium supplemented with 10% FBS, the suspension was added to the cells at a MOI of 200 unless otherwise stated. Chemicals were added to the bacterial suspension if applicable. For transwell infection experiments, bacteria were placed in a TC-insert (Sarstedt) with a pore size of 0.4 μm.

## Mass spectrometry

A549 cell pellets were heated at 95 °C for 10 min in lysis buffer containing 2% SDS, 50 mM Tris-HCl (pH 8), 1 mM tris(2-carboxyethyl) phosphine (TCEP) and 4 mM chloroacetamide. The protein content was precipitated with chloroform/methanol. The protein pellets were then solubilized in 8 M urea and digested with Lys-C and trypsin at 37 °C overnight. The resulting peptides were desalted using Sep-Pak cartridges or C-18 stage tips and labeled with TMT reagents for MS analysis in an Orbitrap Fusion Lumos mass spectrometer. MS data were analyzed with Proteome Discoverer v2.4 (Thermo Fisher Scientific) and subsequently with Perseus.

## Gene ontology enrichment analysis

GO enrichment analysis was carried out using ShinyGO 0.76.2 according to the authors' instructions[72] (http://bioinformatics.sdstate.edu/go/).

## In vivo model of bacterial infection

Groups of 15 BALB/c immunocompetent female mice (body weight 20 ± 2 g) were inoculated i.p. with $3 \times 10^3$ CFU *A. baumannii* (FDA-CDC AR-BANK#0280) suspended in 0.5 mL BHI broth containing 5% mucin. T5224 was dissolved in a vehicle solution at a concentration of 50 mg/mL as suggested in the corresponding patent[73]. Briefly, 500 mg of T5224 was dissolved in a mixture composed of 3.9 mL 0.5 mol/L NaOH and 5.1 mL pure $H_2O$. We then added 1.5 g of Plasdone K29/32 to obtain a 50 mg/mL solution at pH 8.3. Vehicle solution (15 animals) and T5224 (15 animals) were administered orally 23 h before and 1 h after *A. baumannii* infection at a dose of 250 mg/kg. Five mice per group were euthanized 8 hpi, and the lung, spleen, and kidney were harvested for analysis.

## Histology

Formalin-fixed samples were embedded in paraffin and 5-μm sections were prepared using a MICROM HM 355S rotational microtome (Thermo Fisher Scientific). The slides were deparaffinized and rehydrated by sequential immersion in xylene, ethanol and water before staining with hematoxylin and eosin using an H&E staining kit (Abcam).

## Bacterial burden

Mice were sacrificed at 8 hpi. The organs were weighed and homogenized to release the bacteria. CFUs were determined using the dilution plating technique and were divided by the weight of the organs for normalization.

## Heat inactivation of bacteria

Bacteria were washed twice with RPMI and then resuspended in RPMI supplemented with 10% FBS. The suspension was then heated to 95 °C for 10 min. Complete inactivation was confirmed by the absence of growth on LB agar plates after overnight incubation at 37 °C. Equal amounts of live and inactivated bacteria (MOI 200) were used for infection experiments.

## OMV isolation

One liter of overnight bacterial culture was used for OMV isolation. Bacterial cells were removed by centrifugation (4969 g, 30 min, 4 °C) and passing the supernatant through a 0.22-μm PVDF filter. The filtered medium was centrifuged (369548.3 g, 1 h, 4 °C) to pellet the OMVs, which were washed with PBS. The pelleting and washing steps were repeated twice. The final preparation of washed OMVs was then resuspended in 1 mL PBS and sterilized by passing through a 0.22-μm syringe filter. The concentration was determined using a bicinchoninic acid (BCA) assay.

## Western blot

Cell pellets were mixed with SDS lysis buffer (1% SDS in 50 mM Tris-HCl, pH 8) and immediately heated to 95 °C for 10 min to ensure lysis and protein solubilization. Mouse tissues were lysed in RIPA lysis buffer containing proteinase inhibitor by sonication in a Branson Sonifier 250. The protein concentration was determined using the BCA method. For each sample, 10–15 μg protein was loaded on an SDS polyacrylamide gel and fractionated by electrophoresis. Separated

proteins were transferred to PVDF membranes, which were then blocked with 5% non-fat milk powder in PBS containing 0.1% Tween-20 (PBST). The membranes were then incubated sequentially with the primary antibodies and horseradish peroxidase (HRP)-conjugated secondary antibodies, with intervening washes in PBST. Proteins were detected using the WesternBright chemiluminescence reagent (Advansta). Please refer to the "reporting summary" for the list of antibodies and the dilution used.

### LDH release assay
Cell culture supernatant was collected at the indicated time points, and LDH activity was determined using a Cytotoxicity Detection Kit (Roche).

### RNA isolation and qRT-PCR
Total RNA was extracted using the RNeasy Mini kit (Qiagen), and cDNA was prepared using the RevertAid First Strand cDNA Synthesis Kit (Thermo Fisher Scientific). The cDNA was amplified by qPCR using Takyon No ROX SYBR 2X MasterMix blue dTTP (Eurogentec). The relative abundance of the target sequences was calculated using the $\Delta\Delta Ct$ method. GAPDH was used for internal normalization. For the sequences of the primers please refer to the supplementary "Oligonucleotides" file.

### Chromatin immunoprecipitation assay
We used the Pierce Magnetic ChIP Kit (Thermo Fisher Scientific) according to the manufacturer's protocols. Eluted DNA was analyzed by qPCR to determine the relative abundance of the targets as described above.

### Transfection of mammalian cells
Plasmids were introduced into A549 and HEK 293T cells using poly-ethyleneimine as the transfection reagent, whereas Lipofectamine 2000 was used for the transfection of Huh-7 cells. Lipofectamine RNAiMAX was used for the introduction of siRNA into A549 cells. Pre-designed siRNAs (ON-TARGETplus siRNA SMARTPool) were purchased from Horizon Discovery.

### Luciferase reporter assay
On the day before the experiments, cells were transfected with three plasmids: XRE-Luc (pGL4.43[luc2P/XRE/Hygro], Promega), a Renilla luciferase expression vector driven by the thymidine kinase promoter, and pEGFP-C1 (to monitor transfection efficiency). On the day of the experiments, cells were treated with the indicated chemicals and/or OMVs for 3 h before lysis, and luciferase activity was measured using the Dual-Luciferase Reporter Assay System (Promega). Firefly luciferase activity was normalized to the activity of Renilla luciferase.

### Lipid A and bacterial lysate treatment
A549 cells were treated with purified lipid A from *Salmonella enterica* (sigma Aldrich) at a concentration of 15 μg/mL or *A. baumannii* lysate (1:150 diluted) for 3 hours before measuring luciferase activity. To generate cell lysate from *A. baumannii*, strain 19606 was grown in LB medium till OD600 1. Bacteria from 1 mL of the culture were pelleted, washed, and resuspended in 50 μl of PBS. The suspension was then heated at 95 °C for 10 min. Insoluble material was removed using centrifugation.

### Measurement of tryptophan metabolites
Cell pellets (~1 × 10⁶ cells per sample) were lysed in 200 μl methanol extraction buffer (1 mM TCEP, 1 mM ascorbic acid, 0.1% formic acid in 85:15 methanol:water) containing the internal standards 20 μM homotaurine and 20 μM serotonin-d4. Protein precipitates were removed by centrifugation and the supernatant was freeze-dried overnight. For each sample, 70 μl of 200 mM boric acid buffer was added for reconstitution. We then added 10 μl of 6-aminoquinolyl-*N*-hydroxysuccinimidyl carbamate (AQC) to each sample and incubated at 55 °C for 10 min to derivatize amines. Samples were then centrifuged to remove lipids and remaining particulates, and the supernatant was transferred to MS glass vials.

We injected 5 μl of each sample into an Infinity II Bio liquid chromatography system coupled to a 6495 C triple quadrupole mass spectrometer (Agilent Technologies). Tryptophan metabolites were separated on a Zorbax Extend C18 column (RRHD 2.1 ×150 mm, 1.8 μm; Agilent Technologies) at a flow rate of 0.3 mL/min. Metabolites were eluted in a gradient of solvent A (0.1% formic acid in water) and solvent B (0.1% formic acid in acetonitrile) over a period of 19 min: 0 min 1% B, 0–2 min 1% B, 2–9 min 15% B, 9–14 min 30% B, 14–16 min 60% B, 16–17 min 65% B, and 17–19 min 1% B, followed by a 2-min post-time run. Eluted standards were detected in positive ionization dynamic MRM AJS-ESI mode. The gas temperature was set to 290 °C and the gas flow to 20 L/min. The nebulizer was set to 45 psi. The sheath gas flow was set to 11 L/min, with a temperature of 400 °C. The capillary voltage was set at 3800 V with a nozzle voltage of 500 V. The voltages of the High-Pressure RF and Low-Pressure RF were set to 150 and 60 V, respectively.

### Quantification and statistical analysis
GraphPad Prism v8 was used for statistical analysis. Pairs of variables were compared using unpaired and two-tailed Student's *t* tests. Data are represented as means ± standard deviations.

### Reporting summary
Further information on research design is available in the Nature Portfolio Reporting Summary linked to this article.

## Data availability
Source data are provided with this paper. The proteomics data generated in this study have been deposited in the ProteomeXchange Consortium via the PRIDE partner repository[74] partner repository with the dataset identifier PXD053768. Source data are provided with this paper.

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

## Acknowledgements

The authors thank Dr. Andreas Weigert from Goethe University of Frankfurt, Germany for the generous support with the generation of the MDM cells. The mouse infection experiments were carried out by the commercial service provider Pharmacology Discovery Services Taiwan, Ltd. We would also like to thank Julia Bein from Goethe University of Frankfurt, Germany for the technical assistance with the processing of the histological samples. This work was supported by the Deutsche Forschungsgemeinschaft (DFG, German Research Foundation) – Project-ID 259130777 – SFB 1177, the Cluster Project ENABLE funded by the Hessian Ministry for Science and the Leistungszentrum Innovative Therapeutics (TheraNova) funded by the Fraunhofer Society and the Hessian Ministry of Science and Art.

## Author contributions

Conceptualization, C.K. and I.D.; Methodology, C.K., C.P., A.B., M.T., C.M., L.C., S.K., Y.C., R.R., E.G., I.F., and S.G.; Investigation, C.K., C.P., A.B., C.M., L.C., J.M., S.K., and E.G.; Writing – original draft, C.K.; Writing – review & editing, C.K., C.P., M.T., Y.C., M-W.T., S.G., V.K. and I.D.; Funding acquisition, I.D.; Resources, I.F., M-W.T., S.G., V.K. and I.D.; Supervision, I.D.

## Funding

## Competing interests

The authors declare no competing interests.
