## [Peer Review File · Nature Communications]

REVIEWER COMMENTS

Reviewer #1 (Remarks to the Author):

Acinetobacter baumannii outer membrane vesicles have long been shown to induce cytotoxicity and cause host tissue damage. Most *A. baumannii* OMV research has been focused on biogenesis, identification of virulent factors and induction of host immune response. In this manuscript, the authors revealed a novel mechanism of OMV mediated cytotoxicity via modulating host tryptophan metabolism and the transcription factor FOS, an apoptotic cell death regulator. The authors demonstrated their findings are pan-*A. baumannii* events using multiple laboratory and clinical strains and might be *Acinetobacter*-specific by examining two Gram-negative bacteria, *E. coli* and *Pseudomonas aeruginosa*. Overall the study is well designed and the findings significantly advanced our knowledge in *A. baumannii* OMV associated pathogenicity.

The major concern of this report is lacking sufficient *in vivo* data to support author's rationale for developing an anti-FOS based therapy against *Acinetobacter* infection.

Comments related to this issue are:

(1) Since lung histology (Fig. 2i) is the only data to suggest potential benefit of prescribing anti-FOS based therapy, an in-depth assessment and presentation of pathological findings from H&E stained lung sections should be provided to better support the proposed host-targeted therapy.

(2) Please provide rationale for pre-infection T5224 treatment (Fig. 2g) and how does it relate to clinical setting in dealing with *A. baumannii* infection.

(3) Is the mouse infection described in Fig 2g/i a lethal challenge model? Does T5224 treatment improve survival or morbidity (such as weight loss, activity/mobility)?

(4) What are the mouse groups (1, 2 and 3) indicated in Fig. 2i?

Additionally, it would benefit the reader greatly if the authors can provide a graphic summary for their exciting findings.

Reviewer #2 (Remarks to the Author):

The study is focused in the mechanisms of pathogenicity of a bacteria *A. baumannii* that is highly pathogenic and resistant to therapy. The authors show that the cellular toxicity elicited by this bacterium is mediated by the induction of the FOS transcription factor. The induction of the FOS protein is itself mediated by a cascade of events including the activation of the tryptophane-2-dioxygenase (TDO) enzyme leading to increased kynurenine which then activates the Aryl hydrocarbon Receptor, AhR. The activation of TDO appears to rely on outer membrane vesicles components of *A. baumannii*. Most of the experiments are done in cell lines, some are in mice. The authors used inhibitors of FOS, AhR, TDO and in some cases siRNAs. In addition to delineating the mechanisms of action, the implications of the study is to identify putative inhibitors of *A. baumannii* cytotoxicity, which is very relevant taking into account the multidrug resistance of this infection.

All in all, this is a very good study encompassing the discovery of a new pathway of pathogenicity, the delineation of the mechanisms of action and the identification of putative drug candidates (at an early

stage).

There are however some issues that need to be addressed.

1- Much of the work relies on inhibitors of different activities or pathways. This can be very useful as some of these inhibitors could prove to be good drug candidates. However there are specificity issues that are not fully addressed by the authors. The authors should provide evidence for specificity from the literature or from their own work. Two of these inhibitors are discussed in more detail below.

2- T5224 is a critical compound for this study as it has been tested in clinical studies. It is used here as a FOS inhibitor which it is indeed. However the structure (figure 2) includes aromatic components and this type of compound may bind to the AhR, a receptor which displays a large promiscuity in terms of ligand binding. It seems important for the sake of this paper to clearly show that this is not the case since AhR is also involved in the pathway. This could be easily done using the XRE-Luc assay.

3- The same is also relevant for lipid A and for similar reasons.

4- An important issue is the reversibility of the activation of FOS. Obviously the AhR is down regulated, so one wonders how much the effect is sustained. Kinetic studies may be useful here.

5- Some of the gel data shown are relatively poor and the experiments should be repeated. For example the GAPDH bands in figure 4e and 4g. It is even not clear that the different bands are from the same gels in these figures. One wonders how many times these experiments were repeated. Obviously this needs to be addressed by the authors.

Reviewer #3 (Remarks to the Author):

The authors present interesting findings on the potential role of Fos in promoting cell death in *A. baumannii* infected cells or cells exposed to *A. baumannii* OMVs. The differences in in vitro experiments are quite modest – roughly 2 fold for cytotoxicity assays. It is not clear if these modest differences translate into an in vivo impact. The authors need to very significantly improve the in vivo data to obtain a clear understanding of whether fos inhibition provides an in vivo benefit, and if so, if it reduces inflammation and bacterial load, or just one of the two.

Major comments

1) Overall the in vivo data are intriguing but not convincing. There is no quantification in Figure 2i, which is needed. For example, if the authors examined 50 fields of view, how much evidence of inflammation is there with or without the Fos inhibitor? Is there a statistically significant difference? To expand further on this point, a reduction in bacterial load was only observed in roughly half the mice treated with the fos inhibitor. These data should be made more robust with more replicates. If Figure 2i was repeated in “non-responder” mice, would differences in inflammation be observed? Is a reduction in bacterial load not required for in vivo efficacy of the fos inhibitor? If the authors increase the inoculum, would they have a lethal infection model? In this case they could determine the impact of the fos inhibitor on the outcome of infection, and score both bacterial load and inflammation. It is not clear if the fos inhibitor has a robust effect in vivo or if this is very variable and limited to only a subset of mice.

2) Perhaps the discrepant in vivo bacterial load results could be due to an effect of the fos inhibitors on the bacteria directly? Do the fos inhibitors slow the growth rate of the bacteria or reduce their survival in vitro? The impact of fos inhibition on host cell death is modest – often 2 fold or less. Is it possible that host cell death is not the major factor and rather that a direct effect on bacterial survival is the main impact of the fos inhibitors. Each fos inhibitor should be tested.

3) The authors rely on CDC strain 280 for in vivo infections. However, I did not see in vitro data for this strain. Accompanying in vitro experiments would be important to determine if this strain is inducing fos and acting similar to the other strains tested.

4) Figure 5d – Is this very minor difference biologically significant? Maybe acidification does not play an important role?

5) Lines 394-7 – “Surprisingly, OMVs still induce robust lipid A dependent AHR activation in HEK 293T cells (Fig. 5i). This indicates a novel lipid A sensing mechanism independent of TLR4 and caspase 4/5/11, acting upstream of TDO and AHR”

This is not the only possible explanation for the data. I would make this statement less definitive. The next sentence presents an alternative model anyway.

6) Line 146 – “Given that FOS induction is the most potent host response to *A. baumannii* infection”. This is an overstatement based on the response of A549 cells to one strain of *A. baumannii* and using one approach to measure the host response.

7) The authors should similarly be cautious comparing *A. baumannii* to other bacteria (Figure 1e). In each case only one strain is used. The conclusion about this result needs to be qualified by stating that this compared a small number of strains.

Minor comments

1) Figure 1e labeling – 17968 should be 17978?

2) Figure 1c – the y-axis should be set to zero

3) Lines 342-344 – “The use of a highly virulent clinical isolate allowed the infection of immunocompetent animals, closely reflecting the characteristics of *A. baumannii* infection in humans”. Most *A. baumannii* infections in humans are in at least partially immunocompromised patients. I think the authors are trying to highlight that they were able to use immunocompetent mice rather than those rendered neutropenic as is often done. The statement should be corrected, however. In addition the authors relied on mucin in the inoculum to enhance pathogenicity, which is not mimicking natural infection.

We thank the reviewers for their constructive comments and suggestions, which we have addressed by performing additional experiments and analysis. The figures and text have been adjusted accordingly. Below, we respond to the reviewers' comments and criticisms point by point.

REVIEWER COMMENTS

Reviewer #1 (Remarks to the Author):

Acinetobacter baumannii outer membrane vesicles have long been shown to induce cytotoxicity and cause host tissue damage. Most *A. baumannii* OMV research has been focused on biogenesis, identification of virulent factors and induction of host immune response. In this manuscript, the authors revealed a novel mechanism of OMV mediated cytotoxicity via modulating host tryptophan metabolism and the transcription factor FOS, an apoptotic cell death regulator. The authors demonstrated their findings are pan-*A. baumannii* events using multiple laboratory and clinical strains and might be *Acinetobacter*-specific by examining two Gram-negative bacteria, *E. coli* and *Pseudomonas aeruginosa*. Overall the study is well designed and the findings significantly advanced our knowledge in *A. baumannii* OMV associated pathogenicity.

The major concern of this report is lacking sufficient *in vivo* data to support author's rationale for developing an anti-FOS based therapy against *Acinetobacter* infection.

Comments related to this issue are:

(1) Since lung histology (Fig. 2i) is the only data to suggest potential benefit of prescribing anti-FOS based therapy, an in-depth assessment and presentation of pathological findings from H&E stained lung sections should be provided to better support the proposed host-targeted therapy.

To adequately address this valid criticism, we consulted the clinical pathologists of the University Hospital Frankfurt and performed a detailed pathological examination of the tissues (lung, spleen, liver, and kidney) of infected mice treated with the FOS inhibitor T5224 or control (vehicle). This revealed that the differences we initially noted in the lung might not be relevant from a clinical point of view. Therefore, we decided to remove Fig. 2i. However, we did observe an obvious reduction in the accumulation of neutrophils and fibrin on the surface of liver and spleen of the animals treated with T5224, which means that the mice have no or a less severe peritonitis. The red arrows indicate aggregates of neutrophils and fibrin:

Since we infected the animals intraperitoneally, peritonitis is expected. A reduction of neutrophils and fibrin on the serosal surface of these organs suggests that T5224 treatment can reduce peritonitis and inflammation in the *A. baumannii* peritoneal infection model. We added these results in Supplementary Fig. 2.

(2) Please provide rationale for pre-infection T5224 treatment (Fig. 2g) and how does it relate to clinical setting in dealing with *A. baumannii* infection.

The main reason why we chose to do the pre-infection treatment is the rapid progression of the infection in the model we used. In our experience and according to the literature, intraperitoneal infection with *A. baumannii* progresses very rapidly, and animals usually die within a day (please see the kill curve in the response to comment 3). Therefore, treatment prior to infection is needed to keep up with the progression of pathogenesis.

In *A. baumannii* infections in humans, the infection progresses much more slowly, over days. We therefore assume that in humans treatment could well be initiated after the onset of infection. Unfortunately, this is very difficult to model in mice because mice efficiently clear an infection with *A. baumannii*, and many infection models, including pneumonia, actually do not lead to death of the animals. One exemption is the intraperitoneal injection of bacteria, which results in rapid mortality. The development of a mouse model that closely reflects the biology of human infection is still a challenge in the field of *A. baumannii* research. We added this explanation in the Results and Discussion sections.

(3) Is the mouse infection described in Fig 2g/i a lethal challenge model? Does T5224 treatment improve survival or morbidity (such as weight loss, activity/mobility)?

Yes, intraperitoneal infection leads to rapid mortality of the animals. T5224 treatment did not result in a statistically significant improvement in survival ($P=0.1462$). We included this result in Supplementary Fig. 2.

(4) What are the mouse groups (1, 2 and 3) indicated in Fig. 2i? Additionally, it would benefit the reader greatly if the authors can provide a graphic summary for their exciting findings.

Fig. 2i has been deleted to avoid confusion.

Thank you very much for this suggestion. The main results are presented in this graphical summary, which we have added to the manuscript as Supplementary Fig. 7.

Reviewer #2 (Remarks to the Author):

The study is focused in the mechanisms of pathogenicity of a bacteria *A. baumannii* that is highly pathogenic and resistant to therapy. The authors show that the cellular toxicity elicited by this bacterium is mediated by the induction of the FOS transcription factor. The induction of the FOS protein is itself mediated by a cascade of events including the activation of the tryptophane-2-dioxygenase (TDO) enzyme leading to increased kynurenine which then activates the Aryl hydrocarbon Receptor, AhR. The activation of TDO appears to rely on outer membrane vesicles components of *A. baumannii*. Most of the experiments are done in cell lines, some are in mice. The authors used inhibitors of FOS, AhR, TDO and in some cases siRNAs. In addition to delineating the mechanisms of action, the implications of the study is to identify putative inhibitors of *A. baumannii* cytotoxicity, which is very relevant taking into account the multidrug resistance of this infection.

All in all, this is a very good study encompassing the discovery of a new pathway of pathogenicity, the delineation of the mechanisms of action and the identification of putative drug candidates (at an early stage).

There are however some issues that need to be addressed.

1- Much of the work relies on inhibitors of different activities or pathways. This can be very useful as some of these inhibitors could prove to be good drug candidates. However there are specificity issues that are not fully addressed by the authors. The authors should provide evidence for specificity from the literature or from their own work. Two of these inhibitors are discussed in more detail below.

2- T5224 is a critical compound for this study as it has been tested in clinical studies. It is used here as a FOS inhibitor which it is indeed. However the structure (figure 2) includes aromatic components and this type of compound may bind to the AhR, a receptor which displays a large promiscuity in terms of ligand binding. It seems important for the sake of this paper to clearly show that this is not the case since AhR is also involved in the pathway. This could be easily done using the XRE-Luc assay.

Thanks for pointing this out. At the suggestion of the reviewer, we performed the XRE-Luc assay. We did not detect any significant activation of the luciferase reporter by T5224 or SR11302, the two FOS inhibitors we used, whereas the positive control FICZ potently activated the reporter.

We also performed RT-qPCR experiments to measure the levels of *CYP1A1* transcript, one of the major transcriptional targets of AHR. The positive control, FICZ treatment, significantly increased *CYP1A1* transcript levels. SR11302 treatment did not have any effects while T5224 treatment only led to a very small increase of *CYP1A1* transcript levels.

All together, these results do not support T5224 and SR11302 to be relevant inducers of AHR. Moreover, FOS acts downstream of AHR in our model (please refer to the new Supplementary Fig. 7). Even if the AHR processes some promiscuity to the FOS inhibitors,

this should not compromise our conclusion. We added these results in the Supplementary Fig. 5.

3- The same is also relevant for lipid A and for similar reasons.

To address the reviewer's concern, we treated the cells with purified lipid A or lysate from *A. baumannii* bacteria. Interestingly, neither of these treatments induced an activation of the AHR reporter indicating that lipid A alone is not sufficient to activate AHR, but that OMVs are required. OMVs might allow easy internalization into the host cells or contain other bioactive molecules necessary for the activation of AHR. These findings are in good agreement with the results presented in Fig. 5.

We added these data in Supplementary Fig. 5.

4- An important issue is the reversibility of the activation of FOS. Obviously the AhR is down regulated, so one wonders how much the effect is sustained. Kinetic studies may be useful here.

We followed the reviewer's advice and monitored protein levels upon OMV exposure. After 3 hours of OMV treatment, AHR protein levels were severely reduced. Of note, protein levels of FOS and CYP1A1, the two AHR targets, remained significantly elevated even 6 hours after OMV treatment. Thus, induction of AHR targets is maintained even after AHR is degraded. This could potentially be explained by the activity of residual AHR or the stability of the mRNAs of the targets.

We included this result as Supplementary Fig. 4 in the manuscript.

5- Some of the gel data shown are relatively poor and the experiments should be repeated. For example the GAPDH bands in figure 4e and 4g. It is even not clear that the different bands are from the same gels in these figures. One wonders how many times these experiments were repeated. Obviously this needs to be addressed by the authors.

We replaced the problematic panels with data from the other replicates.

Reviewer #3 (Remarks to the Author):

The authors present interesting findings on the potential role of Fos in promoting cell death in *A. baumannii* infected cells or cells exposed to *A. baumannii* OMVs. The differences in in vitro experiments are quite modest – roughly 2 fold for cytotoxicity assays. It is not clear if these modest differences translate into an in vivo impact. The authors need to very significantly improve the in vivo data to obtain a clear understanding of whether fos inhibition provides an in vivo benefit, and if so, if it reduces inflammation and bacterial load, or just one of the two.

Major comments

1) Overall the in vivo data are intriguing but not convincing. There is no quantification in Figure 2i, which is needed. For example, if the authors examined 50 fields of view, how much evidence of inflammation is there with or without the Fos inhibitor? Is there a statistically significant difference? To expand further on this point, a reduction in bacterial load was only observed in roughly half the mice treated with the fos inhibitor. These data should be made more robust with more replicates. If Figure 2i was repeated in “non-responder” mice, would differences in inflammation be observed? Is a reduction in bacterial load not required for in vivo efficacy of the fos inhibitor? If the authors increase the inoculum, would they have a lethal infection model? In this case they could determine the impact of the fos inhibitor on the outcome of infection, and score both bacterial load and inflammation. It is not clear if the fos inhibitor has a robust effect in vivo or if this is very variable and limited to only a subset of mice.

We consulted the clinical pathologists of the University Hospital Frankfurt and performed a detailed pathological examination of the tissues (lung, spleen, liver, and kidney) of infected mice treated with the FOS inhibitor T5224 or control (vehicle). This revealed that the differences we initially noted in the lung might not be relevant from a clinical point of view. Therefore, we decided to remove Fig. 2i.

However, we did observe an obvious reduction in the accumulation of neutrophils and fibrin on the surface of liver and spleen of the animals treated with T5224.

Since we infected the animals intraperitoneally, peritonitis is expected. A reduction of neutrophils on the serosal surface of these organs indicates that T5224 treatment can reduce peritonitis and inflammation in the *A. baumannii* peritoneal infection model. We added these results in Supplementary Fig. 2.

The infection model is a lethal one.

T5224 treatment did not result in a statistically significant improvement in survival ($P=0.1462$). We included this result in Supplementary Fig. 2.

Furthermore, we chose tissues from mice with comparable bacterial load (the higher one) for our biochemical and histological analysis. This means the differences were not due to a difference in bacterial burden. Nevertheless, we repeated the mouse infection experiment using the strain 19606, and we again did not see any significant difference in bacterial loads after T5224 treatment. In conclusion, we believe that T5224 does not affect the bacterial burdens of the animals.

2) Perhaps the discrepant in vivo bacterial load results could be due to an effect of the fos inhibitors on the bacteria directly? Do the fos inhibitors slow the growth rate of the bacteria or reduce their survival in vitro? The impact of fos inhibition on host cell death is modest – often 2 fold or less. Is it possible that host cell death is not the major factor and rather that a direct effect on bacterial survival is the main impact of the fos inhibitors. Each fos inhibitor should be tested.

Thanks for this constructive comment. We tested whether SR11302 or T5224 have any inhibitory effect on the growth of *A. baumannii*. The results were negative indicating that the effects of the FOS inhibitors are not due a change in bacterial survival.

We added these data in Supplementary Fig. 2.

3) The authors rely on CDC strain 280 for in vivo infections. However, I did not see in vitro data for this strain. Accompanying in vitro experiments would be important to determine if this strain is inducing fos and acting similar to the other strains tested.

We repeated some key experiments using strain #0280:

In A549 cells infected with strain #0280, FOS was highly induced. We added this result in Fig. 1.

T5224 treatment alleviated cell death induced by #0280 infection in A549 cells. This result has been included in Fig. 2.

Finally, #0280 infection also resulted in a reduction of AHR protein levels, which is consistent with the results obtained from other strains. This result has been added to Fig. 4.

In conclusion, the #0280 strain acts similarly to the other strains we used in the study.

4) Figure 5d – Is this very minor difference biologically significant? Maybe acidification does not play an important role?

We agree with the reviewer that lysosomal acidification might only play a secondary role. We modified the text accordingly by adding the following sentence in the results section: “Endolysosomal acidification might play a secondary role in the activation of AHR as the effect of bafilomycin A1 was relatively small.”

5) Lines 394-7 – “Surprisingly, OMVs still induce robust lipid A dependent AHR activation in HEK 293T cells (Fig. 5i). This indicates a novel lipid A sensing mechanism independent of TLR4 and caspase 4/5/11, acting upstream of TDO and AHR”

This is not the only possible explanation for the data. I would make this statement less definitive. The next sentence presents an alternative model anyway.

We followed the suggestion of the reviewer and modified the sentence as follows: “This suggests the lipid A-dependent activation of AHR is independent of TLR4 and caspase 4/5/11.”

6) Line 146 – “Given that FOS induction is the most potent host response to *A. baumannii* infection”. This is an overstatement based on the response of A549 cells to one strain of *A. baumannii* and using one approach to measure the host response.

As suggested by the reviewer, we modified the sentence:

“Given that FOS is the most highly induced protein in *A. baumannii* infected cells”

7) The authors should similarly be cautious comparing *A. baumannii* to other bacteria (Figure 1e). In each case only one strain is used. The conclusion about this result needs to be qualified by stating that this compared a small number of strains.

We updated the text according to the reviewer's suggestions.

"*A. baumannii* therefore particularly induced an unusually high level of FOS in the host cells."

was changed to:

"The induction of FOS by *A. baumannii* is therefore more potent than that by *E. coli* and *P. aeruginosa*." (line 138 in the revised manuscript)

"*A. baumannii* infections result in a distinctively strong induction of FOS when compared to other similar bacterial pathogens"

was changed to:

"*A. baumannii* infections result in a stronger induction of FOS when compared to the two other similar bacterial pathogens that we tested" (line 345 in the revised manuscript)

"We found that *A. baumannii* infections cause an atypical upregulation of FOS"

was changed to:

„We found that *A. baumannii* infections cause a high upregulation of FOS" (line 356 in the revised manuscript)

We also added this sentence to the discussion section:

„It should be noted that in the current study we compared only a small number of bacterial strains. In the future, it would be interesting to investigate more bacteria with regard to the induction of FOS and the potential prevention of cytotoxic effects by FOS inhibitors, as these could be used as broad-spectrum host-directed therapeutics for bacterial infections."

Minor comments

1) Figure 1e labeling – 17968 should be 17978?

Right. The error has been corrected.

2) Figure 1c – the y-axis should be set to zero

Figure 1c has no y-axis. Maybe the reviewer is referring to another figure?

3) Lines 342-344 – "The use of a highly virulent clinical isolate allowed the infection of immunocompetent animals, closely reflecting the characteristics of *A. baumannii* infection in humans". Most *A. baumannii* infections in humans are in at least partially immunocompromised patients. I think the authors are trying to highlight that they were able to use immunocompetent mice rather than those rendered neutropenic as is often done. The statement should be corrected, however. In addition the authors relied on mucin in the inoculum to enhance pathogenicity, which is not mimicking natural infection.

We deleted this statement and also modified other parts of the text to avoid any potential overstatement.

Abstract

Original: „Pharmacological inhibition of FOS reduced the pathogenicity of *A. baumannii* in mice and its cytotoxicity in cell-based models specifically.“

Revised: „Pharmacological inhibition of FOS reduced the pathogenicity of *A. baumannii* in cell-based models, and similar results were also observed in a mouse infection model.“

Introduction

Original: „The pharmacological inhibition of FOS specifically prevented cell death induced by *A. baumannii* and suppressed pathogenicity in a mouse model of infection.“

Revised: „The pharmacological inhibition of FOS specifically prevented cell death induced by *A. baumannii* in cell-based models and led to a reduction of severity in a mouse model of infection.“

Discussion:

Original: Given the current antibiotic resistance crisis, we propose host-directed therapies targeting the transcription factor FOS to help turn the tide against infections with multidrug-resistant strains of *A. baumannii*.“

Revised: „Given the current antibiotic resistance crisis, we propose the transcription factor FOS as a promising candidate for the development of host-directed therapies against infections with multidrug-resistant strains of *A. baumannii*.“

We also added this paragraph to the discussion (line 425 – 430):

“This study provides first promising results regarding the reduction of the severity of intraperitoneal *A. baumannii* infection in a mouse model using FOS inhibitors. However, a more thorough investigation is needed to further consolidate the potential of FOS inhibitors as host-directed-therapeutics against bacterial infections. For example, it would be useful to include pneumonia models in future studies, which might be more representative for human infections.“

REVIEWERS' COMMENTS

Reviewer #1 (Remarks to the Author):

The authors have adequately addressed all my comments in the revised manuscript.

Reviewer #2 (Remarks to the Author):

Thank you for addressing the questions I have raised concerning your article. I have no additional questions.

Reviewer #3 (Remarks to the Author):

The main issue continues to be the lack of in vivo efficacy data supporting the use of the inhibitor. The authors now show that the inhibitor does not significantly enhance survival nor reduce bacterial load. In in vitro experiments in A549 cells, there are very minimal if any impacts on cytokine production. The authors now mention liver and spleen neutrophil levels in Supplementary Figure 2h, but these findings are not quantified across many images from many mice and thus no statistics are presented. At present there are no data supporting the in vivo efficacy of the inhibitor. Perhaps the authors can test mediators of inflammation in their in vivo model?

Reviewer #1 (Remarks to the Author):

The authors have adequately addressed all my comments in the revised manuscript.

Thank you so much.

Reviewer #2 (Remarks to the Author):

Thank you for addressing the questions I have raised concerning your article. I have no additional questions.

Thank you so much.

Reviewer #3 (Remarks to the Author):

The main issue continues to be the lack of *in vivo* efficacy data supporting the use of the inhibitor. The authors now show that the inhibitor does not significantly enhance survival nor reduce bacterial load. In *in vitro* experiments in A549 cells, there are very minimal if any impacts on cytokine production. The authors now mention liver and spleen neutrophil levels in Supplementary Figure 2h, but these findings are not quantified across many images from many mice and thus no statistics are presented. At present there are no data supporting the *in vivo* efficacy of the inhibitor. Perhaps the authors can test mediators of inflammation in their *in vivo* model?

Thank you for raising this concern about our *in-vivo* data. We agree that the evidence supporting the use of FOS inhibitors as therapeutics is limited. In response, we have revised the text to better align our claims with the available *in-vivo* data.

1.

We removed the phrases “explore alternative therapeutic strategies” and “representing a promising target for the development of host-targeted therapies”.

2.

“We found that *A. baumannii* infections cause a high upregulation of FOS, whose activity results in host cell death and pathogenesis *in vitro* and *in vivo*”

Is changed to :

“We found that *A. baumannii* infections cause a high upregulation of FOS, whose activity results in host cell death”

3.

“Given the current antibiotic resistance crisis, we propose the transcription factor FOS as a promising candidate for the development of host-directed therapies against infections with multidrug-resistant strains of *A. baumannii*.”

Is changed to :

“Given the current antibiotic resistance crisis, we propose further in vivo investigation of FOS inhibitors in bacterial infections. This represents a direction for developing therapies alternative to traditional antibiotics.”

4.

The following parts are deleted:

“Particularly, the clinical isolate FDA-CDC AR-BANK#0280 used in our mouse infection experiments is resistant to many common drugs except antibiotics of last resort such as carbapenems and colistin. The limited therapeutic options make infections with multidrug-resistant *A. baumannii* difficult to treat, and antibiotics of last resort often have serious side effects. For example, colistin is neurotoxic and nephrotoxic, limiting the dosage and duration of treatment . Excitingly, the FOS inhibitor T5224 mitigates the in vivo toxicity of this challenging infection (Fig. 2g,h, Supplementary Fig. 2h), offering a potential alternative treatment strategy.”

“as these could be used as broad-spectrum host-directed therapeutics for bacterial infections.”

We are also optimizing additional mouse models for *A. baumannii* infection, including lung infection models. These models are more suitable for testing mediators of inflammation, as mice have a slower mortality rate when infected through the lungs. This slower progression allows for an extended time window for physiologically relevant inflammation to develop. However, this aspect is beyond the scope of the current paper.